# Noninvasive Brain Stimulation for Improving Cognitive Deficits and Clinical Symptoms in Attention-Deficit/Hyperactivity Disorder: A Systematic Review and Meta-Analysis

**DOI:** 10.3390/brainsci14121237

**Published:** 2024-12-09

**Authors:** Yao Yin, Xueke Wang, Tingyong Feng

**Affiliations:** 1Faculty of Psychology, Southwest University, Chongqing 400715, China; 2Key Laboratory of Cognition and Personality, Ministry of Education, Southwest University, Chongqing 400715, China

**Keywords:** NIBS, ADHD, cognitive deficits, clinical symptoms

## Abstract

**Objective**: Noninvasive brain stimulation (NIBS) is a promising complementary treatment for attention-deficit/hyperactivity disorder (ADHD). However, its efficacy varies due to diverse participant profiles and methodologies. This meta-analysis, registered with PROSPERO (CRD42023457269), seeks to assess NIBS efficacy in improving cognitive deficits and clinical symptoms in individuals with ADHD. **Methods**: We systematically searched five databases (October 2024) for randomized controlled trials focusing on cognitive functions and clinical symptoms in individuals meeting the DSM/ICD criteria for ADHD. A meta-analytical synthesis was conducted using RevMan 5.4.1. **Results**: Meta-analyses found significant improvement in inhibitory control, working memory, and inattention in active transcranial direct current stimulation (tDCS) groups compared with sham groups. Conversely, repetitive transcranial magnetic stimulation (rTMS) did not demonstrate significant therapeutic benefits for ADHD symptoms. Additionally, four transcranial random noise stimulation (tRNS) and three transcranial alternating current stimulation (tACS) studies demonstrated promising improvements in executive functions and the alleviation of ADHD symptoms. **Conclusions**: The findings from this meta-analysis highlight NIBS as a promising adjunctive therapy for managing ADHD, advancing both theoretical knowledge and practical treatment options in this field.

## 1. Introduction

Attention-deficit/hyperactivity disorder (ADHD) is a persisting and heterogeneous neurodevelopmental disorder with childhood onset, which is characterized by persistent patterns of inattention and/or hyperactivity/impulsivity (clinical/core symptoms) [1,2,3]. ADHD creates substantial burdens and costs, encompassing poor academic outcomes, subpar occupational functioning, and strained relationships and family dynamics [1,4]. Moreover, with a worldwide prevalence of around 5%, a broader range of more effective interventions and prevention for people with ADHD are developing [5].

Above and beyond the core symptoms, individuals with ADHD often experience cognitive deficits, particularly in executive functions, which encompass both “cool” and “hot” cognitive abilities. “Cool” executive functions involve processing abstract and emotionally neutral information, such as inhibitory control, working memory, and cognitive flexibility, which are essential for analytical problem-solving, planning, and reasoning. Conversely, “hot” executive functions are more closely intertwined with emotions and affective processes, ultimately enabling emotional regulation, self-motivation, and impulse control [6,7,8]. Individuals with ADHD frequently display core symptoms and cognitive deficits, with over 90% of them unable to achieve full recovery or sustained remission of these conditions [9]. Hence, delivering interventions at the earliest stage possible for ADHD, based on scientific discoveries and evidence-based approaches, is imperative.

When medication-based treatment strategies work in the treatment of ADHD, the short-term benefit is remarkable. However, they are tolerated by only 50% of patients and are particularly susceptible to adherence problems in adolescence, with side effects that cannot be ignored [10]. In addition, it is essential to note that although cognitive behavioral therapy interventions can target specific executive functioning deficits in individuals with ADHD, their efficiency is limited [11]. Clearly, there is a pressing concern for more effective and less side-effect-prone treatments for ADHD in the long term.

Noninvasive brain stimulation (NIBS) techniques such as transcranial direct current stimulation (tDCS), transcranial random noise stimulation (tRNS), transcranial alternating current stimulation (tACS), and repetitive transcranial magnetic stimulation (rTMS) are of increasing concern as valuable tools in human neuroscience research and treatment of neuropsychiatric disorders, especially those related to executive disorder syndromes. As emerging techniques that may offer long-term therapeutic effects, rTMS and tDCS have been introduced in the treatment of ADHD over the past decade, whereas tACS and tRNS have been incorporated into ADHD treatment in the past three years. A comprehensive and systematic recent meta-analysis observed that tDCS treatment in ADHD has no significant effects on inhibition, processing speed, and inattention [12]. However, an updated systematic review highlighted that when tDCS was targeted at specific sites and administered over multiple sessions, the majority of studies indicated significant therapeutic effects for ADHD intervention [13]. Similarly, the subsequent largest randomized clinical trial of tDCS in adults with ADHD found a significant reduction in inattention symptoms with a large effect size [14]. Likewise, some recent randomized controlled trials (RCTs) have shown promising results, highlighting improvements in inhibitory control and working memory following tDCS treatment for ADHD [13,15,16]. Moreover, there has been growing interest in the use of rTMS, tRNS, and tACS for the treatment of ADHD. Excitingly, these techniques have demonstrated promising behavioral outcomes, effectively improving critical aspects of executive functioning, as well as the core symptoms associated with ADHD [17,18,19]. These findings further suggest that NIBS may hold great potential as a complementary approach to traditional ADHD treatments. Actually, heterogeneity in results has been observed due to variations in participant characteristics, stimulation parameters, outcome measurement methods, and experimental designs. Given the discrepancies in these aspects, whether and how NIBS techniques initiate their effects on cognitive deficits and clinical symptoms in ADHD are still unclear.

To address this question, we conducted this meta-analysis to determine the efficacy of NIBS in improving cognitive deficits and reducing clinical symptoms in individuals with ADHD by conducting a comprehensive and up-to-date meta-analysis. Over the past decade, there has been a significant increase in the number of studies involving rTMS (three studies), tDCS (eighteen studies), tACS (three studies), and tRNS (four studies) for ADHD treatment. According to the outcome measures in these studies, cognitive effects were assessed across distinct domains, such as inhibitory control, working memory, and cognitive flexibility, to elucidate the specific cognitive domains impacted by NIBS. Of note, this study also explored potential moderating variables that could influence intervention outcomes, intending to provide valuable guidance to optimize the intervention process.

## 2. Methods

This study followed the Preferred Reporting Items in Systematic Reviews and Meta-Analyses (PRISMA) guidelines and the research protocol registered in PROSPERO (CRD42023457269).

### 2.1. Inclusion and Exclusion Criteria

We used the P.I.C.O.S. (Population, Interventions, Comparators, Outcomes) framework to identify relevant studies. The inclusion criteria were as follows: (1) for the types of studies, only the randomized controlled trials (RCTs) studies were included; (2) for the types of participants, children and adults with ADHD were diagnosed by physicians or other qualified mental health professionals following the Diagnostic and Statistical Manual of Mental Disorders, Fourth or Fifth Edition (DSM-IV/DSM-V), or the International Classification of diseases (ICD-10); (3) for the types of intervention, tDCS, tACS, tRNS, and rTMS were applied in ADHD; (4) for the types of comparators, the electric montages of the sham group were identical to those of the active group; and (5) for the types of outcome measures, we were interested in the clinical measures of ADHD symptoms and performance measures on cognitive tasks. The exclusion criteria were as follows: (1) the intervention combined cognitive training and noninvasive brain stimulation, (2) the studies were not in English, and (3) the studies were not peer reviewed and not published.

### 2.2. Search Strategy

Two authors independently conducted a comprehensive search on multiple databases, including PubMed, Web of Science, PsycInfo, PsycoArticle, and EBSQ, from inception to the end of October 2024. Any disagreements were resolved by the corresponding author. Our search strategy involved combining subject terms (“attention deficit hyperactivity disorder”, “hyperkinetic disorder”, “ADHD”, “attention”, “inattention”, “hyperactivity”, or “impulsivity”) and intervention terms (“noninvasive brain stimulation”, “NIBS”, “transcranial electric stimulation”, “tES”, “transcranial direct current stimulation”, “tDCS”, “transcranial alternating current stimulation”, “tACS”, “transcranial random noise stimulation”, “tRNS”, “transcranial magnetic stimulation”, or “TMS”). In addition, we performed an extensive search of the references cited in the selected studies, as well as earlier meta-analyses studies and systematic reviews, to identify any additional relevant studies that could be included in our analysis.

### 2.3. Study Selection and Data Extraction

After eliminating duplicates, the first author examined the titles and abstracts using a machine learning tool, ASReview [20], and the validity of ASReview was also verified [21,22]. After labeling three relevant and three irrelevant studies, ASReview captured the features of the relevant studies and ranked all retrieved articles by their relevance. As a result, 58 most relevant studies were exported, and the first author took 10% of the excluded studies to determine if the tool made the right decisions [22]. Subsequently, a final list of the included studies was compiled through a consensus process by the first author and the corresponding author based on what was deemed relevant. Following this, they independently exacted the pertinent data from included studies: the study information (i.e., last name of the first author, the year of publication, and the study design), characteristics of the participants (e.g., the mean age, sample size, and current medication), intervention characteristics (e.g., sessions, timing, and electric montages), and outcomes (i.e., the means and standard deviations of the post-tests of cognitive task performance or clinical measures for the active and sham groups).

### 2.4. Assessment of Risk of Bias

Two reviewers independently assessed the risk of bias using the Cochrane Collaboration’s risk of bias tool (RoB2) [23]. The RoB2 rates the risk of bias across six domains, namely, selection bias, performance bias, detection bias, attrition bias, reporting bias, and other biases. Every study was evaluated and classified as low, high, or some concerns of risk of bias for each domain.

### 2.5. Statistical Analysis

Review Manager (RevMan) version 5.4.1 was used for statistical analysis in the current study. To investigate the effectiveness of noninvasive brain stimulation in addressing cognitive deficits and clinical symptoms in individuals with ADHD and to mitigate enormous heterogeneity, we categorized cognitive outcome measures into three domains: inhibitory control, working memory, and cognitive flexibility. Similarly, clinical symptom measures were classified into two domains: inattention and hyperactivity/impulsivity. Inhibitory control in the included studies was measured by the commission errors in the go/no-go task (GNG) and the flanker task, the accuracy or stop signal reaction time in the stop-signal task (SST), and the error response in the Stroop task. For working memory measures, we included the number of correct or correct rates in the N-back task and the accuracy in the digit span-backward test. We contained the perseverative errors in the Wisconsin card sorting test (WCST) for cognitive flexibility measures. Furthermore, for inattention, we included the omission errors, commission errors, and reaction time in the continuous performance test (CPT); the commission errors in the visual attention test (VAT); the inattention subscale scores on the Adult ADHD Self-Report Scale (ASRS); and the inattention subscale scores on the German Adaptive Diagnostic Checklist for ADHD (FBB-ADHD). As for the hyperactivity/impulsivity measures, we included hyperactivity/impulsivity subscale scores on the ASRS and FBB-ADHD and the number of spacebar presses in the CPT [24]. Of note, all the outcomes above were measured as continuous variables.

As the outcomes were from different tasks or scales, we estimated the effect sizes using the standardized mean differences (SMD) with a 95% confidence interval (CI). The SMD is based on Cohen’s d and can be interpreted using the following thresholds: 0.2 represents a small effect, 0.5 represents a medium effect, and 0.8 represents a large effect. To enhance the generalizability of the study findings to different contexts [25] and lend greater weight to the study with a small sample size [26], a random effects model was employed in this study. Furthermore, the evaluation of heterogeneity was facilitated by utilizing the I^2^ statistic, denoting the proportion of the variance between studies in the overall variance. Notably, a confluence of I^2^ values of 25%, 50%, and 75% connotes low, moderate, and high heterogeneity, respectively.

Furthermore, to gain deeper insights into the determinants of intervention efficacy, we conducted subgroup analyses on specific outcomes, focusing specifically on trials with an adequate sample size. By adopting this approach, we aimed to discern the underlying factors that contributed to the effectiveness of interventions in a more comprehensive and nuanced manner.

### 2.6. Sensitivity Analyses

Sensitivity analysis is crucial in meta-analysis to assess the robustness and reliability of the findings. In this study, we employed the “leave-one-out” method, a widely used approach for sensitivity analysis [27]. The “leave-one-out” method systematically removes each study from the meta-analysis and reanalyzes the combined effect size. By iteratively excluding one study at a time and recalculating the effect estimate, it allows us to evaluate each study’s influence on the overall outcome.

### 2.7. Publication Bias Assessment

To assess the publication bias in our meta-analysis, we utilized funnel plots using RevMan 5.4.1 and conducted Egger’s test using Matlab 2021b, which provided a comprehensive evaluation. Funnel plots visually represent the relationship between effect sizes and their standard errors, allowing for the identification of potential publication bias. However, when using funnel plots to assess publication bias, it is generally recommended to include a minimum of 10 studies [28]. Additionally, Egger’s test employs statistical techniques to quantify the degree of asymmetry in the funnel plot and assess the significance of the bias. Importantly, when the funnel plot shows symmetrical and concentrated points above the midline, and the intercept of Egger’s test approaches zero with a *p*-value greater than 0.05, it indicates a low level of publication bias.

## 3. Results

### 3.1. Study Selection

According to the PRISMA flowchart (Figure 1), our search strategy initially yielded 2392 citations. After removing duplicates (476 studies), we used ASReview to screen the remaining 1916 studies by reviewing the titles and abstracts. After excluding the irrelevant literature, we carefully assessed 65 full-text studies for eligibility. Ultimately, we identified 18 tDCS studies and 3 rTMS studies that met our inclusion criteria [14,15,16,17,24,29,30,31,32,33,34,35,36,37,38,39,40,41,42,43,44]. Of note, due to the limited outcome measures of included studies, we were unable to conduct a meta-analysis on the clinical effects or cognitive effects of tACS (three studies) [45,46,47] and tRNS studies (four studies) [18,48,49,50]. Consequently, our meta-analysis focused solely on the effects of tDCS and rTMS studies in ADHD. The general characteristics of the included studies are briefly summarized in Table 1 (tDCS), Table 2 (rTMS), and Table 3 (tRNS and tACS), providing an overview of crucial details such as the study design, sample size, intervention type, duration, and primary outcome measures.

### 3.2. Risk of Bias

For tDCS studies, a comprehensive evaluation of the risk of bias is presented in Figure 2. Overall, most studies demonstrated a low risk of bias in most domains, except performance bias. Specifically, a considerable proportion of studies were needed to provide adequate details regarding the blinding process in performance bias. Additionally, a small number of studies indicated a high risk of bias in the domains of selection (*n* = 1), performance (*n* = 1), reporting (*n* = 3), and other biases (*n* = 1). Notably, the results of sensitivity analyses convincingly indicated that the inclusion of high-risk studies did not significantly impact the overall findings and conclusions of this meta-analysis (further details can be found in Section 3.4).

Regarding rTMS studies, it is noteworthy that the included studies exhibited a low risk of bias across all domains. Furthermore, among the various domains assessed, only one study showed a high risk of bias, specifically in the domain of other biases. This high risk was attributed to baseline differences between the active group and the sham group. Figure 3 summarizes the risk of bias analysis conducted for the included studies. In addition, all the studies of tRNS and tACS exhibited no high-risk bias across all domains (Figure 4). The justification for the judgment was also elucidated in Appendix A.

### 3.3. tDCS Effectiveness

#### 3.3.1. Inhibitory Control

A total of 17 trials from 12 studies involving 582 patients were included to evaluate the efficacy of tDCS in improving inhibitory control among individuals with ADHD. The analysis of cognitive data from the participants revealed that tDCS had a significant positive effect on inhibitory control performance compared with the sham group (Table 4; Figure 5A; *SMD* = −0.21, 95% *CI*: [−0.39, −0.04], *p* = 0.02). Subgroup analyses further indicated that tDCS intervention enhanced inhibitory control in specific populations within the ADHD cohort (Appendix A). Notably, significant or marginal improvements in inhibitory control were observed when the study participants were children (*SMD* = −0.19, 95% *CI*: [−0.40, 0.02], *p* = 0.07), with a male proportion of less than 50% (*SMD* = −0.58, 95% *CI*: [−1.03, −0.14], *p* = 0.01), or when they were medicated before tDCS stimulation (*SMD* = −0.27, 95% *CI*: [−0.53, 0.00], *p* = 0.05). Moreover, targeted stimulation of the F3 (left dorsolateral prefrontal cortex, dlPFC) or F4 (right dlPFC) regions (*SMD* = −0.17, 95% *CI*: [−0.36, 0.02], *p* = 0.08), a stimulation duration of 15 min (*SMD* = −0.37, 95% *CI*: [−0.67, −0.06], *p* = 0.02), one session stimulation (*SMD* = −0.2, 95% *CI*: [−0.39, −0.02], *p* = 0.03), and an offline stimulation timing (*SMD* = −0.41, 95% *CI*: [−0.72, −0.10], *p* = 0.01) were associated with enhanced inhibitory control in individuals with ADHD. Finally, when the experimental design involved a crossover approach (*SMD* = −0.23, 95% *CI*: [−0.43, −0.04], *p* = 0.02) or a single-blind system (*SMD* = −0.18, 95% *CI*: [−0.37, −0.01], *p* = 0.07), the inhibitory control in individuals with ADHD was also found to be improved. All the subgroup results are represented in Appendix A.

#### 3.3.2. Working Memory

Our analysis comprised a total of 12 trials from nine studies involving 390 patients. Cognitive data obtained from the participants demonstrated a statistically significant positive impact of tDCS on working memory performance compared with the sham group (Table 4; Figure 5B; *SMD* = 0.31, 95% *CI*: [0.03, 0.59], *p* = 0.03). Subgroup analyses provided compelling evidence that one session stimulation (*SMD* = 0.43, 95% *CI*: [0.06, 0.81], *p* = 0.02), a stimulation duration of 20 min (*SMD* = 0.22, 95% *CI*: [0.00, 0.45], *p* = 0.05), and online stimulation timing (*SMD* = 0.29, 95% *CI*: [0.05, 0.53], *p* = 0.02) were associated with improved working memory in individuals with ADHD (Appendix A). Additionally, experimental designs involving a crossover approach (*SMD* = 0.35, 95% *CI*: [0.04, 0.67], *p* = 0.03) also resulted in enhanced working memory in individuals with ADHD.

#### 3.3.3. Cognitive Flexibility

This analysis encompassed a sample size of 94 patients across four trials from two studies. Examining the cognitive data obtained from these participants, we found no statistically significant impact of tDCS on cognitive flexibility performance when compared with the sham group (Table 4; Figure 5C; *SMD* = −0.61, 95% *CI*: [−1.48, 0.25], *p* = 0.17).

#### 3.3.4. Inattention

Our comprehensive analysis comprised eight trials from seven studies, including a total of 277 patients. The results revealed a statistically significant impact of tDCS on inattention performance compared with the sham group (Table 4; Figure 5D; *SMD* = −0.66, 95% *CI*: [−1.33, 0.00], *p* = 0.05). Further subgroup analyses unveiled encouraging findings, demonstrating that tDCS intervention exhibited promising effectiveness in ameliorating inattention, specifically in adults (*SMD* = −1.03, 95% *CI*: [−1.71, −0.34], *p* = 0.003) with ADHD (Appendix A). Moreover, targeted stimulation of the F3 or F4 regions (*SMD* = −0.66, 95% *CI*: [−1.33, 0.00], *p* = 0.05), offline stimulation timing (*SMD* = −0.84, 95% *CI*: [−1.59, −0.10], *p* = 0.03), and multi-session stimulation (*SMD* = −0.77, 95% *CI*: [−1.53, −0.01], *p* = 0.05) were associated with notable improvements in inattention among individuals with ADHD. Furthermore, studies employing a parallel design (*SMD* = −0.93, 95% *CI*: [−1.6, −0.25], *p* = 0.007) or a single-blind design (*SMD* = −1.31, 95% *CI*: [−2.59, −0.04], *p* = 0.04) showed improved working memory among individuals with ADHD.

#### 3.3.5. Hyperactivity/Impulsivity

This analysis included a sample size of 94 patients in four trials from four studies. When analyzing the clinical measure data collected from these participants, we did not observe a statistically significant effect of tDCS on hyperactivity/impulsivity compared with the sham group (Table 4; Figure 5E; *SMD* = −0.41, 95% *CI*: [−1.05, 0.23], *p* = 0.21). It is important to note that we did not conduct further subgroup analyses for these measures, due to the limited number of trials available for cognitive flexibility and hyperactivity/impulsivity.

### 3.4. rTMS Effectiveness

In this analysis, we examined data from 137 patients across four trials from three studies to investigate the impact of rTMS on core symptoms of ADHD, specifically, inattention and hyperactivity/impulsivity. However, our findings did not reveal any statistically significant effects of rTMS compared with the sham group (Table 4; Figure 6; *SMD* = 0.04, 95% *CI*: [−0.48, 0.55], *p* = 0.89) in alleviating these symptoms. Of note, due to the limited number of papers available, further subgroup analyses and a meta-analysis on the effects of rTMS on cognitive deficits in ADHD were not conducted.

### 3.5. Publication Bias and Sensitivity Analysis

With a limited number of studies available, only funnel plots were used to assess the intervention effects of tDCS on inhibitory control and working memory in ADHD (Appendix A). The funnel plots and statistical tests showed no evidence of publication bias in inhibitory control (Egger’s test *p* = 0.8034) and working memory (Egger’s test *p* = 0.3458). Furthermore, Egger’s tests of cognitive flexibility (*p* = 0.3242), inattention (*p* = 0.5437), and hyperactivity/impulsivity (*p* = 0.6384) suggested that no publication bias was observed. As for the rTMS studies, Egger’s test of clinical symptoms suggested that no publication bias was observed (*p* = 0.6088). Sensitivity analysis showed that regardless of whether each study on tDCS or rTMS interventions was sequentially excluded, the significance of the meta-analysis results remained unchanged (see details in Appendix A).

## 4. Discussion

This meta-analysis aimed to investigate the effectiveness of NIBS in mitigating cognitive deficits and clinical symptoms in individuals with ADHD. The meta-analysis results revealed that tDCS exhibited a significant enhancing effect on inhibitory control, working memory, and attention in ADHD patients. Additionally, rTMS did not demonstrate a significant improvement in the core symptoms of ADHD. However, due to a limited number of relevant studies and the various outcomes in these studies, we did not conduct a meta-analysis on tACS and tRNS. Nevertheless, these studies indicated a trend of improvement in cognitive deficits and core symptoms of ADHD after applying tACS or tRNS. For more detailed descriptions, refer to Section 4.3. Furthermore, we conducted subgroup analyses to explore further the moderating variables that influenced intervention effects if the studies were sufficient (see details in Appendix A).

### 4.1. tDCS Effectiveness

Currently, people are increasingly recognizing that the significance of ADHD for public health is far more severe than commonly perceived [51]. As a result, there is a strong demand for interventions targeting ADHD, and tDCS has increasingly gained favor among psychiatrists and physicians due to its high safety profile, minimal side effects, and lower cost compared with medication-based treatments. This is also why tDCS composes the majority of the retrieved NIBS literature.

In the present study, we observed a notable improvement in the effect size of tDCS treatment on cognitive control, working memory, and attention in individuals with ADHD. tDCS involves placing two or more electrodes on the scalp of the participant and delivering a weak electrical current to modulate neural plasticity [52]. Advances in neuroimaging, neurophysiology, and computational neuroscience have led to new insights in psychiatry, suggesting that psychiatric disorders, including ADHD, are caused by functional disruptions in distributed networks within the brain [53]. For example, meta-analyses have found hyperconnectivity between the frontoparietal network (FPN), default mode network (DMN), and affective network (AN) in ADHD, while they have found hypoconnectivity between the FPN and the ventral attention network (VAN) and somatosensory network (SSN) [54]. As they are the core regions of the FPN, brain stimulation of the dlPFC and inferior frontal cortex (IFC) can indirectly downregulate brain areas within the DMN [55]. This is why F3/F4 (dlPFC) and Fp2/F8 (right IFC) are commonly used and effective targets for tDCS intervention in ADHD. Similarly, studies on tDCS have found that stimulating the right IFC activates the target area and influences the prefrontal-basal ganglia circuitry to improve inhibitory control [56]. Therefore, from the perspective of modulating brain networks or circuits, the potential of tDCS in treating ADHD should not be underestimated.

Compared with the deactivation of the dlPFC, the functional impairment of the right IFC in cognitive tasks is more specific to ADHD. However, subgroup analyses have shown that stimulating F3/F4 significantly improves working memory and inattention in ADHD, while stimulating other targets, including F8/Fp2, does not improve any cognitive functions or core symptoms of ADHD. This may be because the largest current density used in experiments targeting the IFC, which is deeper in the brain compared with the dlPFC, is just 5 A/m^2^, indicating that a higher current density is needed to yield effective results [10]. Additionally, subgroup analysis showed significant improvements in inhibitory control in ADHD when tDCS was administered without discontinuing medication before or during the treatment period. Conversely, when ADHD patients discontinued medication both before and during the treatment period, tDCS significantly improved working memory in ADHD. Therefore, the effects of the drug may confound the results of tDCS, which should be carefully considered when analyzing and applying the findings of this meta-analysis. Regarding experimental design, regardless of whether tDCS was administered online or offline, in a crossover or parallel fashion, and in a double-blind or single-blind manner, varying degrees of improvement in cognitive deficits and core symptoms of ADHD were observed with tDCS. However, perhaps due to insufficient research or less sensitive measurement indicators, we did not observe improvements in cognitive flexibility and hyperactivity/impulsivity following tDCS in ADHD patients. Based on these findings, in future scientific research and clinical practice of tDCS for ADHD, it may be beneficial to target ADHD-specific regions, such as the IFC, with higher current densities. Furthermore, as tDCS effects are cumulative, multiple treatment sessions are also recommended.

### 4.2. rTMS Effectiveness

rTMS can alter neural activity, modulate cortical excitability, induce localized changes in brain activity, and promote plasticity at the network level [30,57]. The main advantage of rTMS compared with tDCS is its precise spatial targeting, which typically leads to better and faster therapeutic effects. However, the number of studies using rTMS for the treatment of ADHD is fewer than those using tDCS. This could be attributed to the fact that rTMS is a suprathreshold stimulation with a 0.03% risk of inducing seizures. For instance, in a study investigating rTMS for ADHD, it was reported that one participant had a seizure after three sessions and was unable to continue with the study [30]. Moreover, in addition to the fact that the equipment itself is expensive and not portable, the maintenance of its equipment is also costly. For example, rTMS devices require a cooling apparatus, which adds to the cost compared with tDCS.

Nonetheless, our meta-analysis revealed significant improvements in core symptoms of ADHD with rTMS. By delivering brief and intense pulse currents to coils placed on the head using electromagnetic induction, rTMS generates electric fields in the brain [53]. The potential therapeutic effects of rTMS are theoretically based on neurostimulation of specific brain regions that exhibit functional deficits corresponding to different cognitive functions. Research has indicated that ADHD patients’ executive dysfunction and core symptomatology are associated with abnormalities in the cortico-striato-thalamo-cortical circuitry, primarily involving the prefrontal cortex (PFC) and its associated networks [58,59]. Consequently, all the rTMS studies included in our meta-analysis employed high-frequency (>5 Hz) stimulation to activate the PFC to alleviate cognitive impairments and core symptoms in individuals with ADHD.

In recent years, the role of non-high-order networks, such as the somatomotor network (SMN), in various psychiatric disorders has been frequently reported but has not received much attention [60,61]. For instance, a data-driven study based on multivariate prediction has demonstrated that the SMN serves as a transdiagnostic hub across four psychiatric disorders, including ADHD, and has proposed that alterations in somatomotor processes impact symptoms, cognitive functions, and personality [62]. Furthermore, rTMS combined magnetic resonance imaging could significantly improve its therapeutic outcomes in psychiatric disorders [63]. Therefore, in future rTMS research, combining neuroimaging techniques like magnetic resonance imaging with targeted stimulation of SMN cortical regions, including the central posterior gyrus and supplementary motor area, alongside frontal areas, may yield more promising outcomes.

### 4.3. tRNS and tACS Effectiveness

tRNS and tACS have only been used for the treatment of ADHD in the past four years. Due to a limited number of studies and the various outcomes, a meta-analysis of the efficacy of these two NIBS techniques in ADHD has not been conducted. Like tDCS, tACS involves placing two or more electrodes on the patient’s scalp, but tACS delivers alternating current at a specified frequency range (0.01–200 Hz), whereas tDCS operates at a frequency of 0. Due to its current characteristics, tACS has been shown to modulate endogenous brain oscillations [64]. Studies have indicated that ADHD patients exhibit a sustained decrease in P300 amplitude (an electrophysiological marker related to attention) during cognitive tasks such as go/no-go tasks [65]. Based on this, a randomized, single-blind, parallel study used tACS to modulate the P300 amplitude in adults with ADHD [45], aiming to improve their cognitive deficits. The results of this study revealed that compared with the sham group, the active group showed a significant increase in P300 amplitude after 20 min of tACS. Additionally, the active group demonstrated a significant reduction in omission errors during the go/no-go task, indicating improved inhibitory control in individuals with ADHD. Another randomized, single-blind, crossover study suggested that 15 min of tACS in children with ADHD improved risk decision-making abilities in the active group compared with the sham group [47]. However, a recent randomized, single-blind, crossover study targeting Cz and Oz in 15 adults with ADHD found no significant improvement in core ADHD symptoms following a single session of tACS treatment [46]. This may be due to the short washout period between active and sham treatments, which was only one day and likely insufficient.

Unlike tDCS and tACS, tRNS delivers random noise stimulation, and the simultaneous enhancement of cortical excitability can be achieved by using two or more electrodes in tRNS. A randomized, double-blind, crossover study combined tDCS and tRNS with cognitive training to treat ADHD [48,49]. In ADHD children, during five sessions of tDCS stimulation combined with cognitive training, the anodal electrode was placed at F3, and the cathodal electrode was placed at F4. After a one-week washout period, during five sessions of tRNS stimulation combined with cognitive training, the two electrodes were placed at F3 and F4 (bilateral dlPFC). The results revealed that compared with tDCS treatment, tRNS therapy improved the core symptoms, working memory, and processing speed. However, this study had two limitations. First, it did not include a sham group, which meant there was no control group to compare the effects of tRNS treatment. Second, the study combined cognitive training with tRNS, which confounded the treatment outcomes attributed to brain stimulation alone. Therefore, caution should be exercised when applying the results of this study. Furthermore, in a recent randomized, double-blind, parallel study [18], tRNS combined with cognitive training was investigated as an intervention for ADHD children. During ten sessions of tRNS combined with cognitive training, the electrodes were placed at positions F3 and F8. The results revealed that the active tRNS combined with the cognitive training group showed a lower aperiodic exponent, suggesting improved cognitive abilities compared with the sham tRNS combined with the cognitive training group. However, no significant difference was observed between the two groups regarding cognitive task performance. Furthermore, a recent randomized, single-blind, crossover study found that tRNS treatment targeting F3/Fps significantly improved both “cool” and “hot” executive functions in children [50].

In conclusion, the use of tACS and tRNS in the treatment of ADHD is still in its early stages. While some studies have confirmed their potential to improve cognitive deficits and core symptoms of ADHD, more research is needed to establish the effectiveness of tACS and tRNS in ADHD treatment and gain a comprehensive understanding of their mechanisms of action and optimal treatment protocols.

## 5. Limitation

First, due to the more systematic concern and measurement directed toward the “cool” executive functions in NIBS therapy for ADHD, our meta-analysis and discussion focused predominantly on these outcomes. Accordingly, results about “hot” executive functions, being relatively scant, were omitted from the present meta-analysis and discussion. In addition, it is crucial to acknowledge limitations in the longitudinal tracking of NIBS effects. Only three studies conducted follow-up assessments on the efficiency of NIBS, with substantial disparities in the tracking durations—seven days [43], four months [32], and six months [14]. Hence, this study did not conduct a meta-analysis or discuss the long-term efficiency of NIBS for ADHD due to the substantial variability in follow-up durations, compounded by the absence of follow-up data in the referenced Leffa’s study.

## 6. Further Directions

In the future scientific research and clinical practice of NIBS for treating ADHD, it is crucial to ensure more precise targets and standardized stimulation patterns to advance the efficacy of interventions. Current studies predominantly focus on stimulating the dlPFC; however, research indicates that the IFC serves as a more specific biomarker for ADHD compared with the dlPFC [10]. Furthermore, stimulation of the IFC activates the frontal-subcortical circuit, enhancing cognitive abilities such as inhibitory control [56]. Therefore, future research should emphasize the precision of target selection to achieve more targeted therapeutic interventions. Regarding standardized stimulation patterns, accurate measurement of stimulation intensity is essential. For instance, in tDCS treatment, reporting current intensity rather than mere current size is imperative. Consistency in reporting current intensity becomes especially crucial when employing the same current size for treating ADHD patients using different electrode sizes, particularly in the context of high-precision and conventional instruments. Notably, considering that the IFC is located in a deeper brain region than the dlPFC, stimulation of the IFC may require the use of higher current intensities. Additionally, due to the cumulative effects of neuroregulation, we strongly recommend interventions spanning multiple sessions. On this basis, conducting extended studies to assess the prolonged impact and maintenance effects of NIBS on individuals with ADHD is imperative. Therefore, combining more precise target selection with effective stimulation patterns involving varying intensities for different targets, multiple sessions, and diligent tracking may represent a promising avenue for future NIBS interventions in ADHD.

Given the heterogeneity of ADHD, personalized treatment and combination therapies will be pivotal. Tailoring stimulation parameters, target selection, and treatment schedules based on each patient’s neurobiological differences and symptomatology holds the promise of enhancing treatment efficacy, minimizing adverse reactions, and better addressing individual needs. While current evidence shows that NIBS alone has not achieved large effect sizes in ADHD interventions, its potential can be significantly enhanced through the optimization of parameters and combination with other therapies. Moreover, the implementation of combination therapies can overcome the limitations of singular treatment approaches. This approach is particularly valuable in accommodating inter-individual differences among patients as combination therapies afford greater flexibility to meet diverse treatment requirements. The pursuit of personalized precision treatments and the advancement of comprehensive combination therapies represent crucial frontiers in addressing the intricacies of ADHD, offering prospects for superior therapeutic outcomes.

Finally, a deeper exploration of the brain mechanisms behind NIBS intervention in ADHD is necessary. Strengthening research on the neural mechanisms of NIBS in individuals with ADHD, especially at the level of brain networks, will deepen the understanding of the treatment’s foundation and provide theoretical support for devising more precise therapeutic approaches.

## 7. Conclusions

Our study emphasized the potential efficacy of noninvasive brain stimulation (NIBS) in the treatment of attention-deficit/hyperactivity disorder (ADHD). Based on the results of meta-analyses, we found significant improvements in inhibitory control, working memory, and inattention with tDCS for ADHD. Subgroup analysis demonstrated that tDCS showed significant effects in improving these executive functions and symptoms when targeting the F3/F4 regions. To further assess the efficacy of tDCS, future research should explore the effects of tDCS after medication cessation and investigate additional target regions that address the heterogeneity of ADHD. Our findings indicated that rTMS did not significantly improve the core symptoms of ADHD. However, we acknowledge the limited number of studies related to rTMS. Thus, further research is needed to evaluate and establish the reliable efficacy of rTMS in the treatment of ADHD. Additionally, preliminary studies on tRNS and tACS suggested their potential efficacy in addressing the cognitive deficits and core symptoms of ADHD. However, due to limitations in experimental design and a need for more studies, more rigorous research is required to validate their efficacy and determine optimal application methods. By validating the potential efficacy of NIBS treatment in ameliorating ADHD cognitive deficits and symptoms, this study establishes a fundamental basis for advancing further research and facilitating the development of innovative therapeutic strategies, thereby expanding the repertoire of treatment options and instilling increased hope for individuals with ADHD.

## Figures and Tables

**Figure 1 brainsci-14-01237-f001:**
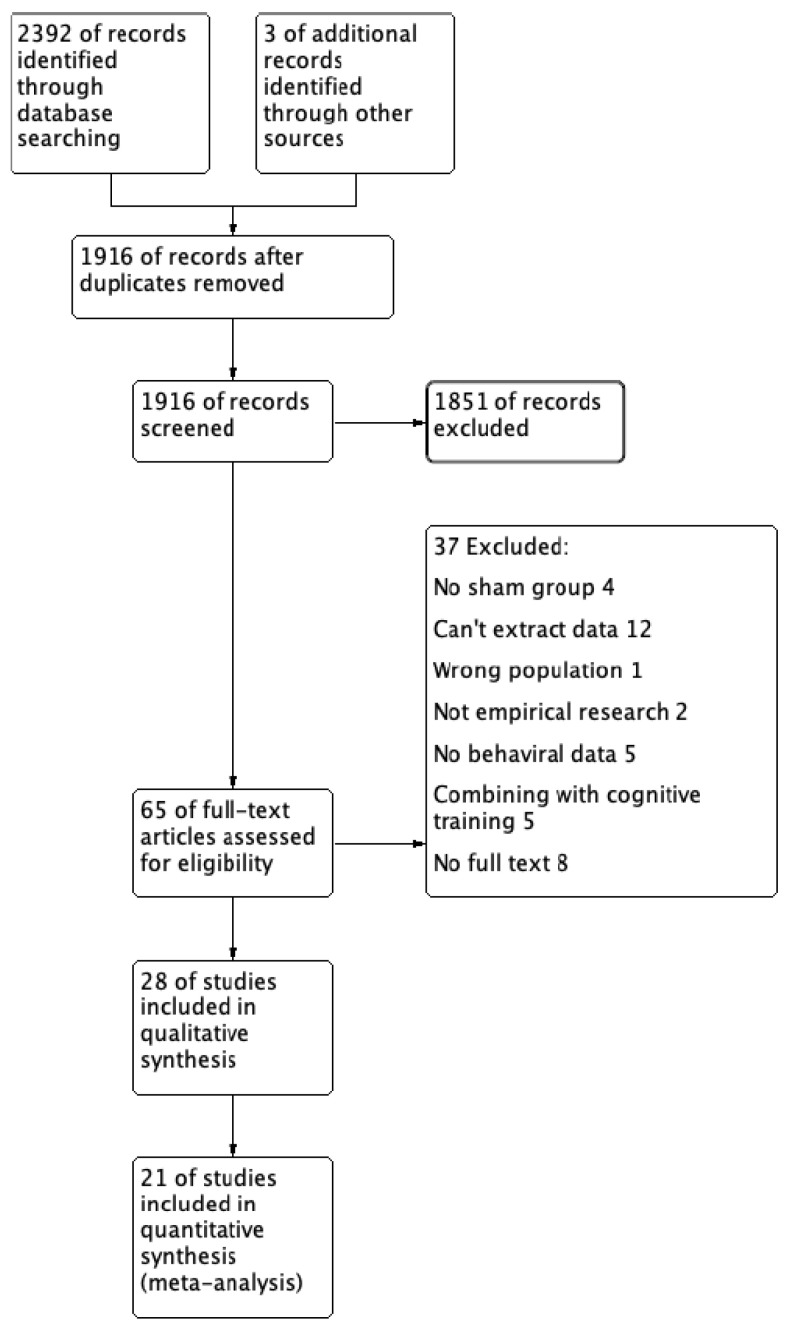
PRISMA diagram of identifying eligible studies.

**Figure 2 brainsci-14-01237-f002:**
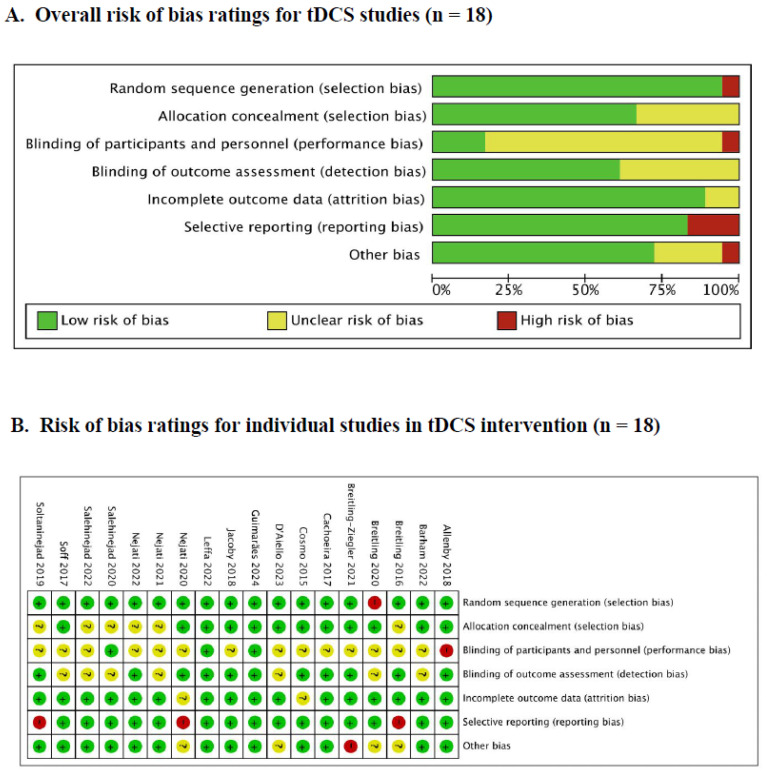
The risk bias of tDCS studies [13,14,15,16,19,24,31,32,33,34,37,38,39,41,43,44].

**Figure 3 brainsci-14-01237-f003:**
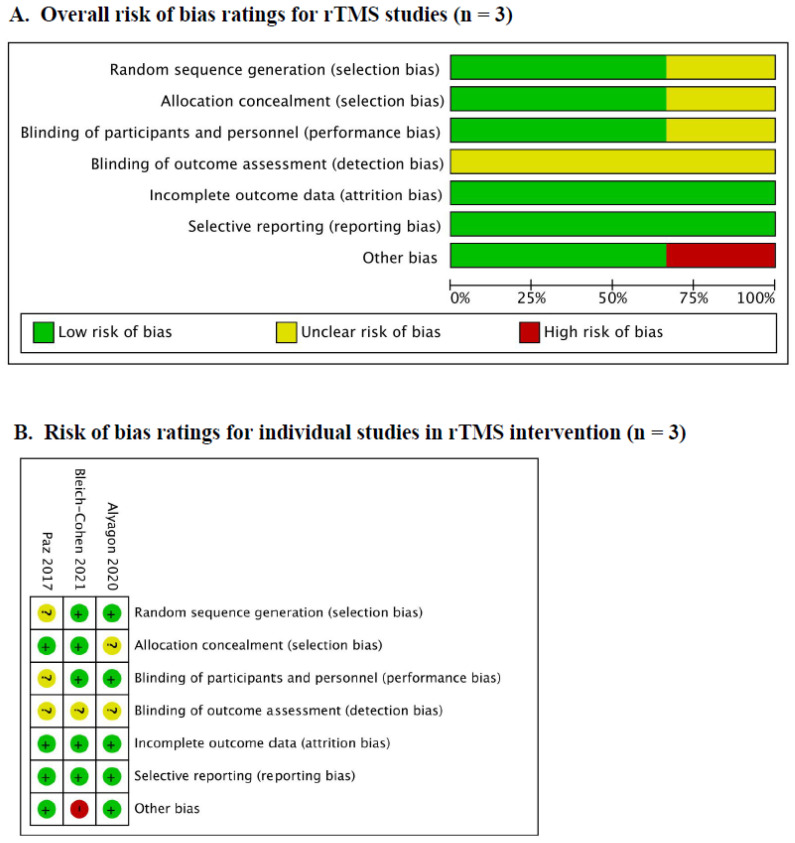
The risk bias of rTMS studies [17,30,40].

**Figure 4 brainsci-14-01237-f004:**
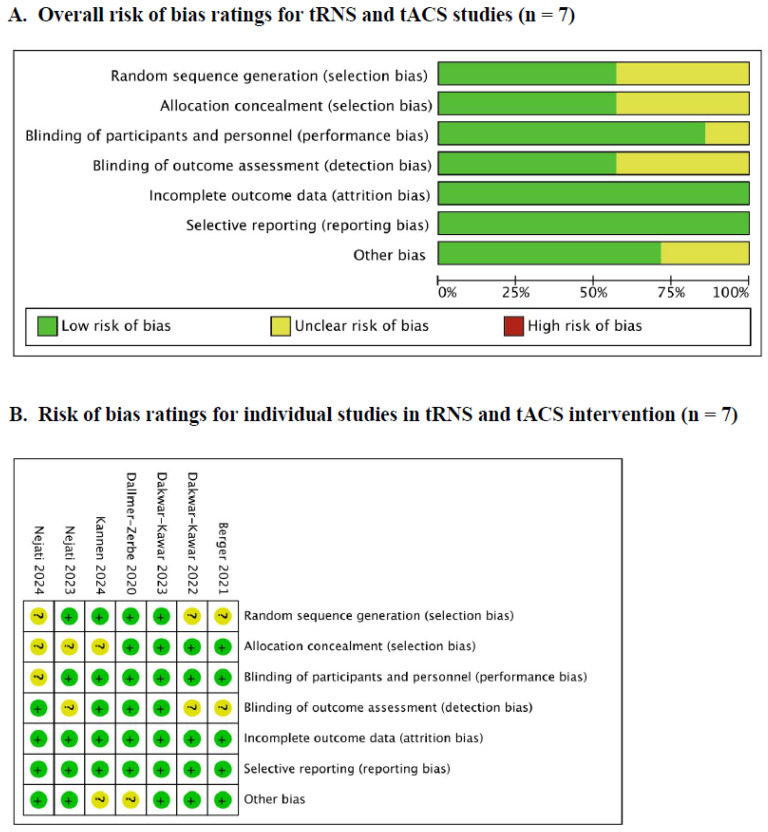
The risk bias of tACS and tRNS studies [18,19,45,46,48,49,50].

**Figure 5 brainsci-14-01237-f005:**
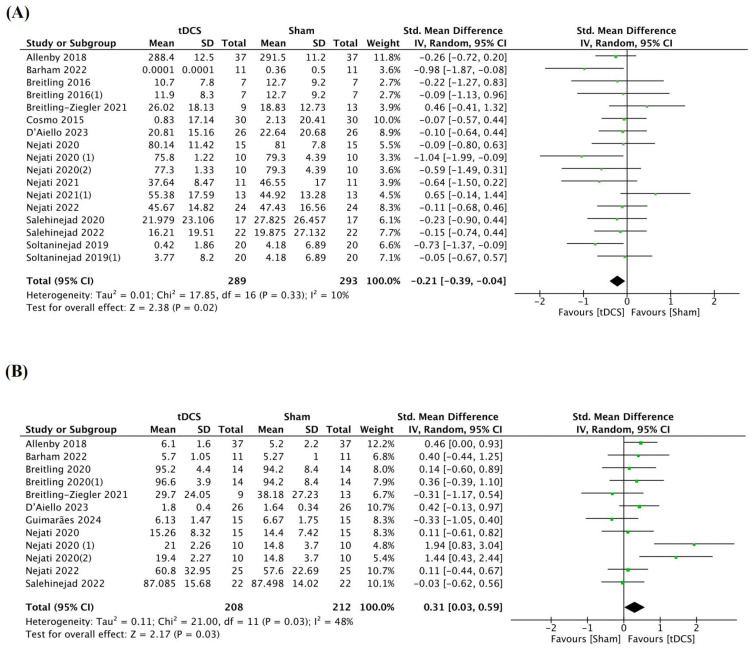
Meta−analysis of measures of (**A**) inhibitory control, (**B**) working memory, (**C**) cognitive flexibility, (**D**) inattention, and (**E**) hyperactivity/impulsivity in tDCS studies [13,14,15,16,19,24,31,32,33,34,37,38,39,41,43,44].

**Figure 6 brainsci-14-01237-f006:**
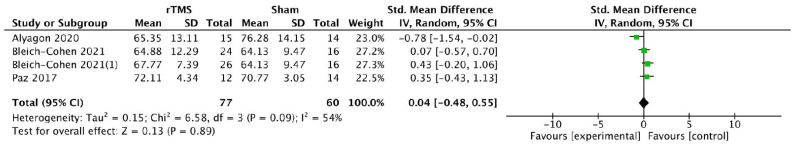
Meta-analysis of measures of core symptoms in rTMS studies [17,30,40].

**Table 1 brainsci-14-01237-t001:** tDCS studies.

		Population				Intervention					Outcomes	
Studies	Design	N	Age (M ± SD)	Male (%)	Drug Naive	Anode/Cathode	Current Intensity (A/m^2^)	Session	Duration (min)	Timing	Clinical	Cognitive
Cosmo 2015 [34]	Randomized Single blind Sham controlled Parallel	Active: 30 Sham: 30	Active: 31.83 ± 11.55 Sham: 32.67 ± 10.37	Active: 56.67 Sham: 60.0	No	F3/F4	0.29	1	20	Offline		Go/no-go task
Breitling 2016 [31]	Randomized Double blind Sham controlled Crossover	7	13.30 ± 1.90	100	Yes	F8/back of the head back of the head/F8	0.29	1	20	Online		Flanker
Cachoeira 2017 [33]	Randomized Single blind Sham controlled Parallel	Active: 9 Sham: 8	Active: 31 ± 6.17 Sham: 33.75 ± 3.65	47.06	No	F4/F3	0.57	5	20	Offline	Adult ADHD Self-Report Scale symptom (ASRS)	
Soff 2017 [43]	Randomized Double blind Sham controlled Crossover	15	14.2 ± 1.2	80	Yes	F3/vertex	1	5	20	Online	Parents’ version of a German Adaptive Diagnostic Checklist for ADHD (FBB-ADHD)	
Allenby 2018 [29]	Randomized Single blind Sham controlled Crossover	37	n/a	74.29	n/a	F3/right- supraorbital area	0.80	3	20	Online		Stop-signal task N-back task Continuous performance test
Jacoby 2018 [24]	Randomized Single blind Sham controlled Crossover	20	22.75 ± 2.80	45	Yes	F3 and F4/below the inion	2	1	20	Offline		Continuous performance test
Soltaninejad 2019 [44]	Randomized Single blind Sham controlled Crossover	20	16.06 ± 0.98	n/a	n/a	Fp2/F3	0.43	1	15	Offline		Go/no-go task
Breitling 2020 [16]	Randomized Double blind Sham controlled Crossover	14	13.30 ± 1.90	86.67	Yes	F8/contralateral supra-orbital area	5	1	20	Online		N-back task
Nejati 2020 [37]	Randomized Double blind Sham controlled Crossover	10	9 ± 1.80	50	Yes	F3/F4 Fp2/F3 F3/Fp2	0.40	1	15	Offline		Go/no-go task N-bask task Wisconsin card sorting test
Salehinejad 2020 [41]	Randomized Single blind Sham controlled Crossover	17	9.53 ± 1.50	70.59	Yes	P4/left shoulder	0.40	1	23	Online		Go/no-go task Shifting attention test
Breitling-Ziegler 2021 [32]	Randomized Double blind Sham controlled Parallel	Active: 9 Sham: 13	Active: 13.22 ± 2.39 Sham: 13.54 ± 1.45	86.36	Yes	IFC/(n/a)	5	5	20	Online		Go/no-go task N-back task
Nejati 2021 [38]	Randomized Single blind Sham controlled Crossover	24	9.25 ± 1.53	63.63	Yes	F3/F4	0.40	1	20	Online		Go/no-go task
Barham 2022 [15]	Randomized Double blind Sham controlled Parallel	Active: 11 Sham: 11	22 ± 2.77	31.8	Yes	F4/F3	0.57	5	20	Offline		Stroop task Digit span task Continuous performance test
Leffa 2022 [14]	Randomized Double blind Sham controlled Parallel	Active: 32 Sham: 32	38.3 ± 9.6	53.12	No	F4/F3	0.57	28	30	Offline	ASRS	
Nejati 2022 [39]	Randomized Single blind Sham controlled Crossover	24	9.25 ± 1.53	66.67	Yes	F4/left forearm	0.40	1	20	Offline		Go/no-go N-back
Salehinejad 2022 [13]	Randomized Single blind Sham controlled Crossover	11	8.86 ± 1.80	50	Yes	F4 and F3/shoulder	0.94	1	15	Online		Go/no-go task N-back Wisconsin card sorting test
D’Aiello 2023 [35]	Randomized Single blind Sham controlled Crossover	26	10.63 ± 1.41	92.3	Yes	F3/Fp2	0.4	1	20	Online		Stop-signal task N-back
Guimarães 2024 [36]	Randomized Triple blind Sham controlled Crossover	15	11.2 ± 3.0	66.67	Yes	F3/contralateral supra-orbital area	0.57	5	30	Offline		Visual attention test Digit span test Inhibiting response test

**Table 2 brainsci-14-01237-t002:** rTMS studies.

		Population				Intervention				Outcomes	
Studies	Design	N	Age (M ± SD)	Male (%)	Drug Naive	Targets	Frequency (Hz)	Sessions	Timing	Clinical	Cognitive
Paz 2018 [40]	Randomized Double blind Sham controlled Parallel	Active: 12 Sham: 14	31.60 ± 6.55	53.85	Yes	Bilateral PFC	18	20	n/a	Conners’ Adult ADHD Rating Scale (CAARS)	
Alyagon 2020 [30]	Randomized Single blind Sham controlled Parallel	Active: 15 Sham: 14	Active: 26.62 ± 0.66 Sham: 27.64 ± 1.58	69.77	Yes	Right PFC	18	15	Offline	CAARS	
Bleich-Cohen 2021 [17]	Randomized Double blind Sham controlled Parallel	Active1: 24 Active2: 26 Sham: 16	Active1: 35.60 ± 8.70 Active2: 35.10 ± 10 Sham: 34.70 ± 9.20	64.52	Yes	Right PFC Left PFC	18	15	Offline	CAARS	

**Table 3 brainsci-14-01237-t003:** tRNS and tACS studies.

		Population				Intervention						Outcomes	
Studies	Design	N	Age (M ± SD)	(%)	Drug Naive	Targets	Current (mA)	Frequency (Hz)	Sessions	Timing	Duration (min)	Clinical	Cognitive
Berger 2021 [48]	Randomized Double blind Active controlled Crossover	19	8.95 ± 1.86	94.74	Yes	F3/F8	0.75	100–640 Hz	5	Offline	20	ADHD-RS diagnostic questionnaire	Digit span test MOXO-CPT task
Dakwar-Kawar 2022 [49]	Randomized Double blind Active controlled Crossover	19	8.95 ± 1.86	94.74	Yes	F3/F8	0.75	100–640 Hz	5	Offline	20		MOXO-CPT task
Dakwar-Kawar 2023 [18]	Randomized Double blind Sham controlled Parallel	Active 11 Sham: 12	Active: 9.25 ± 1.42 Sham: 8.64 ± 1.43	86.96	Yes	F3/F8	0.75	100–640 Hz	10	Offline	20	ADHD-RS diagnostic questionnaire	Digit span test MOXO-CPT task
Nejati 2024 [50]	Randomized Single blind Sham controlled Crossover	18	9.89 ± 1.9	66.67	Yes	F3/Fp2	1	100–640 Hz	1	Online	20		Circle tracing task Go/no-go task 1-back test Wisconsin card sorting task Balloon analogue risk task (BART)
Dallmer-Zerbe 2020 [45]	Randomized Single blind Sham controlled Parallel	Active 9 Sham: 9	31.3 ± 9.89	38.89	Yes	C3, C4, CP3, CP4, P3, P4; T7, T8, TP7, TP8, P7, P8	1	0.5–20 Hz	1	Online	20		Go/no-go task
Nejati 2023 [19]	Randomized Single blind Sham controlled Crossover	18	8.55 ± 1.54	n/a	Yes	F3/Fp2	1.5	n/a	5	Online	15		Balloon analogue risk task
Kannen 2024 [46]	Randomized Single blind Sham controlled Crossover	15	32.53 ± 11.07	73.33	Yes	Cz/Oz	1.5	Active: 9.63 ± 0.69; Sham: 9.67 ± 0.98	1	Offline	18		CPT

**Table 4 brainsci-14-01237-t004:** The brief results of meta-analysis in tDCS and rTMS.

EF/Symptoms	Technique			
	tDCS		rTMS	
	SMD	*p*-Value	SMD	*p*-Value
Inhibitory control	−0.21	0.02 *	-	-
Working memory	0.31	0.03 *	-	-
Cognitive flexibility	−0.61	0.17	-	-
Inattention	−0.66	0.05 *	0.04	0.89
Hyperactivity/impulsivity	−0.41	0.21		

* *p* < 0.05.

## Data Availability

If there is a reasonable justification, the Matlab code utilized for Egger’s test in this study can be obtained from the corresponding author.

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
