# Peer review of "Noninvasive Brain Stimulation for Improving Cognitive Deficits and Clinical Symptoms in Attention-Deficit/Hyperactivity Disorder: A Systematic Review and Meta-Analysis"

_brainsci, 2024, doi:10.3390/brainsci14121237_

Round 1

Reviewer 1 Report

Comments and Suggestions for Authors

[Brain Sciences] Manuscript ID: brainsci-3345253, Yao Yin et al.

Title: Noninvasive brain stimulation for improving cognitive deficits and clinical symptoms in attention deficit hyperactivity disorder: a systematic review and meta-analysis

Noninvasive brain stimulation (NIBS) is a promising complementary treatment for Attention-Deficit/Hyperactivity Disorder (ADHD). In this paper, meta-analysis was conducted to assess NIBS efficacy in improving cognitive deficits and clinical symptoms in individuals with ADHD. The result showed that except repetitive transcranial magnetic stimulation (rTMS), transcranial direct current stimulation (tDCS), transcranial random noise stimulation (tRNS) and transcranial alternating current stimulation (tACS)studies were effective to alleviate ADHD symptoms, suggesting that NIBS may be a promising adjunctive therapy for managing ADHD.

Given a skeptical view, rTMS may not stimulate neurons, while other NIBS methods, including tDCS, tRNS, and tACS, might cause non-specific stimulation.               However, I agree that exploring NIBS as a therapy of ADHD might be potentially important. Furthermore, the manuscript is well-written, clear and important.  

Reviewer 2 Report

Comments and Suggestions for Authors

In my opinion the manuscript is well written, scientifically precise and addresses an important topic in the psychiatric field (that of ADHD and new methodologies to reduce its symptoms). I included the only comment to make the interpretation of the data easier in the previous review.

- A table summarizing the observed effects of each technique on different ADHD symptoms might be beneficial

Reviewer 3 Report

Comments and Suggestions for Authors

Thank you for the opportunity to review this manuscript.

The authors present a well-conducted systematic review and metanalysis that will be of interest to prospective readers. They have included a wider range of therapies in their literature evaluation than previous works, and while some therapies are found not to have a basis strong enough to be evaluated, this observation itself is an added value to the manuscript.

There are two areas where the manuscript can be improved, however.

The authors frame their investigation on the cost of ADHD to society, and there is some discussion regarding the cost of equipment to conduct treatment. However, the manuscript mostly focuses on the analysis of whether the effect of therapy is statistically significant. There is no consistent presentation of the size of the effect of the analyzed therapies, and also no full discussion on what the clinical demands on a therapy outcome would have to show to be considered an effective treatment considering the resources required and alternative treatments.

A minor suggestion is also to ensure that the scales of the subfigures of Figure 5 are consistent in referring to control/ sham or experimental / treatment-favored outcomes.
